# Multiple Glycation Sites in Blood Plasma Proteins as an Integrated Biomarker of Type 2 Diabetes Mellitus

**DOI:** 10.3390/ijms20092329

**Published:** 2019-05-10

**Authors:** Alena Soboleva, Gregory Mavropulo-Stolyarenko, Tatiana Karonova, Domenika Thieme, Wolfgang Hoehenwarter, Christian Ihling, Vasily Stefanov, Tatiana Grishina, Andrej Frolov

**Affiliations:** 1Department of Biochemistry, St. Petersburg State University, 199034 Saint Petersburg, Russia; st021585@student.spbu.ru (A.S.); gm2124@mail.ru (G.M.-S.); vastef@mail.ru (V.S.); tgrishina@mail.ru (T.G.); 2Department of Bioorganic Chemistry, Leibniz Institute of Plant Biochemistry, D-06120 Halle (Saale), Germany; 3Almazov National Medical Research Centre, 197341 Saint Petersburg, Russia; karonova@mail.ru; 4Department of Faculty Therapy, The First Pavlov St. Petersburg State Medical University, 197022 Saint Petersburg, Russia; 5Proteome Analytics Research Group, Leibniz Institute of Plant Biochemistry, D-06120 Halle (Saale), Germany; Domenika.Thieme@ipb-halle.de (D.T.); Wolfgang.Hoehenwarter@ipb-halle.de (W.H.); 6Institute of Pharmacy, Martin Luther University of Halle-Wittenberg, D-06120 Halle (Saale), Germany; christian.ihling@pharmazie.uni-halle.de

**Keywords:** Amadori compounds, biomarkers, glycation, glycation sites, label-free quantification, linear discriminant analysis, mass spectrometry, plasma proteins, type 2 diabetes mellitus

## Abstract

Type 2 diabetes mellitus (T2DM) is one of the most widely spread metabolic diseases. Because of its asymptomatic onset and slow development, early diagnosis and adequate glycaemic control are the prerequisites for successful T2DM therapy. In this context, individual amino acid residues might be sensitive indicators of alterations in blood glycation levels. Moreover, due to a large variation in the half-life times of plasma proteins, a generalized biomarker, based on multiple glycation sites, might provide comprehensive control of the glycemic status across any desired time span. Therefore, here, we address the patterns of glycation sites in highly-abundant blood plasma proteins of T2DM patients and corresponding age- and gender-matched controls by comprehensive liquid chromatography-mass spectrometry (LC-MS). The analysis revealed 42 lysyl residues, significantly upregulated under hyperglycemic conditions. Thereby, for 32 glycation sites, biomarker behavior was demonstrated here for the first time. The differentially glycated lysines represented nine plasma proteins with half-lives from 2 to 21 days, giving access to an integrated biomarker based on multiple protein-specific Amadori peptides. The validation of this biomarker relied on linear discriminant analysis (LDA) with random sub-sampling of the training set and leave-one-out cross-validation (LOOCV), which resulted in an accuracy, specificity, and sensitivity of 92%, 100%, and 85%, respectively.

## 1. Introduction

Diabetes is an ubiquitously spread disease with a worldwide occurrence exceeding 387 million in 2014, and expected to reach 592 million by the year 2035 [1]. Among the total number of cases, more than 90% are represented by the type 2 diabetes mellitus (T2DM), characterized with impaired insulin action and/or insulin secretion [2]. As the onset of the early metabolic alterations (insulin tolerance and hyperglycaemia) is asymptomatic, vascular diabetic complications are often revealed at the moment of T2DM diagnosis [3]. Hence, early discovery of T2DM, or even preceding changes in glucose metabolism, would essentially increase therapy efficiency and reduce the costs required for the treatment of diabetic complications [4]. Obviously, even minor changes in blood glucose concentrations might affect the rates of blood protein glycation [5], i.e., interaction of glucose with free amino groups of proteins, yielding fructosylamines, also known as Amadori compounds [6] (Figure 1). Their further degradation results in the formation of a highly heterogenic group of advanced glycation end products (AGEs) [7], known for their pro-inflammatory impact in the development of vascular diabetic complications [8,9]. However, although AGEs are well-established biomarkers of diabetic complications [10], Amadori compounds might suit better for glycemic control and early diagnostics of T2DM [11].

Fasting plasma glucose (FPG) is the preferred diagnostic parameter, but an increase of blood glucose is also detected in patients suffering from other diseases besides diabetes mellitus [12]. Currently, the blood content of hemoglobin isoform, HbA_1c_, glycated by the N-terminus of its α-chain (˃6.5% of the total hemoglobin fraction), is one of the principle diagnostic criteria of this disease [13] and an efficient tool of long-term (60–90 days) glycemic control [14]. However, HbA_1c_ does not deliver any information about short-term alterations in plasma glucose concentrations accompanying the onset of metabolic syndrome [15]. In contrast, short-living plasma proteins provide a good possibility to decrease the time dimensions of glycemic control. Thus, human serum albumin (HSA), the major blood plasma protein with a half-life of 21 days, can be used as a marker of T2DM [16]. Its global glycation rates can be quantitatively assigned by an array of enzymatic [17], colorimetric [18], immunochemical [19], electrophoretic [20], and chromatographic [21] methods. Importantly, the levels of HSA glycation vary from 1% and 10% in healthy individuals to 20% to 90% in patients with diabetes [22]. However, as the glycation rates at individual lysyl residues in the HSA molecule differ essentially [23], the sensitivity of the potential glycation sites to short-term changes of blood glucose levels might also be different.

In this context, the monitoring of glycation rates at specific glycation sites might be advantageous in comparison to the quantification of global glycation levels. Therefore during the last decade, mass spectrometry was intensively employed in the establishment of such techniques [24,25,26]. In all cases, analytics relied on the bottom-up proteomic approach, based on nano-scaled liquid chromatography-mass spectrometry (nanoLC-MS) and tandem mass spectrometry (MS/MS). Thus, we showed that only some HSA lysyl residues are differentially glycated in plasma of T2DM patients, whereas glycation rates at other potential modification sites seemed to not be affected [27]. Hence, tryptic peptides, representing differentially glycated sites, might be considered as prospective T2DM biomarkers. For some of them, it was additionally confirmed by absolute quantification using internal standardization with dabsylated bi-labeled Amadori-modified peptides [28] or ^13^C, ^15^N synthetic analogs of specifically glycated peptides [29]. As was shown recently, one of the plasma glycation sites (namely K_141_ in haptoglobin) might provide an additional diagnostic tool in combination with well-established T2DM markers, such as fasting plasma glucose (FPG) and HbA_1c_. The main advantage of combining two markers—HbA_1c_ and glycated haptoglobin, K_141_—is the simultaneous consideration of two proteins with different half-life times, i.e., 3 to 4 and 2 to 4 days, respectively. It makes this biomarker is sensitive to long- and short-term fluctuations of blood glucose concentrations. The set of glycated K_141_ of haptoglobin and HbA_1c_ provided a sensitivity of 94%, a specificity of 98%, and an accuracy of 96% to identify T2DM [30].

In this context, it is logical to assume that a biomarker strategy, based on multiple specific glycation sites in plasma proteins, could essentially increase the efficiency of glycemic control and disease prediction. Indeed, the involvement of several glycated proteins with different half-life times (τ_1/2_) allows several time segments in the glycemic control to be addressed without additional analyses. Moreover, this approach might decrease the impact of individual glycation sites in the overall dispersion of the data, when larger cohorts are considered. Therefore, here, we present a mass spectrometry-based biomarker approach relying on multiple glycation sites. We demonstrate its applicability to different time points in the span of glycemic control. Based on the results of linear discriminant analysis (LDA) performed for the cohorts of T2DM patients and individuals without diabetes, we characterize a set of Amadori-modified peptides with a diagnosis accuracy of up to 92%.

## 2. Results

### 2.1. Normalization by Plasma Protein Contents and Tryptic Digestion

According to the applied workflow (Figure 2), the normalization of analyte abundances relied on the determination of plasma protein contents and quality controls. The plasma protein concentrations, determined by the Bradford assay, varied from 36.2 to 82.4 mg/mL (Appendix A), and were 58.8 ± 9.85 and 61.9 ± 13.5 mg/mL in the T2DM and normoglycemic groups, respectively. These values were used for the normalization of protein amounts taken for tryptic digestion and were considered in the label-free quantification procedure (Figure 2). Hence, as an adequate normalization is a pre-requisite for the satisfactory precision of label-free quantification [31], the plasma protein contents, determined by the Bradford assay, were cross-verified by sodium dodecyl sulfate polyacrylamide gel electrophoresis (SDS-PAGE) (protein load of 5 µg per line) (Appendix A), and total line densities were recorded (Appendix A). The whole-line densitometric analysis revealed the average density of 19,483 ± 1289 arbitrary units (AU, relative standard deviation (RSD) = 6.6%), thereby providing sufficient precision for reliable quantification of individual glycated peptides by the signal intensities of corresponding quasi-molecular ions. The following tryptic digestion was considered to be quantitative, as the human serum albumine (HSA) band could not be detected in the SDS-PAGE experiments with tryptic digests (Appendix A), indicating a digestion efficiency better than 95% [27].

### 2.2. Annotation of Prospectively Glycated Peptides

Due to the high resolution and mass accuracy of the measurements (typically more than 12,000 and less than 3 ppm, respectively), individual glycated peptides could be reliably annotated by their retention times (t_R_) and exact masses (*m/z*) of multiple-charged quasi-molecular ions. This step relied on the list of glycated tryptic peptides, obtained from the plasma proteins of T2DM patients, based on the comprehensive works of Zhang et al. [26,32] and representing, to the best of our knowledge, the most complete set of reliably identified in vivo plasma glycation sites (more than 350). Therefore, for each glycated peptide identified by Zhang et al., all possible charge states were calculated with consideration of their amino acid composition. On the basis of this data, characteristic extracted ion chromatograms (XICs) were constructed in the mass ranges of *m/z* ± 0.02 for all matched signals with *z* ≥ 2. For this, the time-of-flight (TOF) scans obtained with pooled diabetic plasma protein digests (*n* = 3) were used. Most of the glycated peptides were annotated with sub-ppm mass accuracies and in all cases represented the minimal mass deviations from the theoretically predicted values in the ranges used for building the corresponding XICs (Table 1). This strategy revealed a total of 51 Amadori-modified peptides present in the pooled plasma protein digests in each sample, representing 10 major plasma proteins. As these features could be reliably detected, i.e., demonstrated a signal to noise ratio ≥ 3 in the corresponding XICs, their relative abundances in the T2DM and without diabetes cohorts were addressed by the label free quantification approach.

### 2.3. Label-free Quantification

A paired Mann-Whitney test performed for the intensities of individual peptide signals (expressed as integral peak areas of corresponding XICs at defined retention times (t_R_s)) revealed 42 differentially (*p* ≤ 0.01) glycated peptides representing nine plasma proteins (Figure 3 and Appendix A). Among this number, for 10 peptides, the biomarker properties were already proposed earlier [27,30], whereas for 32 species, a statistically significant T2DM-related abundance increase was observed here for the first time. Thereby, the levels of glycation at 25 HSA lysyl residues (represented by 27 tryptic peptides and comprising 43% of the total 58 present in the protein sequence) were significantly increased in the plasma of T2DM patients in comparison to the healthy controls. In contrast, only six modification sites were up-regulated in serotransferrin (six tryptic peptides, 10.3% of the totally available lysines), whereas the other proteins were represented with only two (α-2-macroglobulin and Ig kappa chain C region) or one site, demonstrating enhanced glycation in T2DM plasma (Table 1). Most of the annotated prospective biomarkers could distinguish T2DM patients with sensitivities of 85% to 100% and accuracies exceeding 80% (Table 1). Thus, each of the differentially glycated proteins was represented with at least one marker peptide with biomarker sensitivities typically higher than 85% (with one exception for the apolipoprotein A-derived peptide, LAEYHAKATEHLSTLSEK, modified at lysine 219, Table 1). Interestingly, the τ_1/2_ values of these proteins completely covered a time span of three weeks (Table 2).

Integration of selected peptide signals in the quality control (QC) samples revealed high intra- and inter-dayprecision of analyte retention times (t_R_s) and abundances. Thus, the RSD for intra- and inter-dayprecision of t_R_s typically did not exceed 0.54% and 0.94%, whereas for peak areas, it was not higher than 11.0% and 16.46%, respectively (Appendix A).

### 2.4. Sequence Assessment of Differentially Abundant Glycated Peptides

All features demonstrating significantly higher abundances in patients with diabetes were identified by tandem mass spectrometry (MS/MS, Appendix A). The MS/MS spectra of 18 glycated peptides could be acquired in data-dependent acquisition (DDA) experiments with tryptic digests obtained from pooled T2DM plasma (*n* = 3). The fragmentation of the other 24 species was achieved in targeted LC-MS/MS experiments. The sequences of the annotated peptides were unambiguously confirmed by a search against Uniprot human database (downloaded 7 June 2016) and verified by manual interpretation of the MS/MS spectra as exemplified in Figure 3 (for the spectral information see Appendix A). The sequences of six peptides (3, 4, 32, 34, 37, 40) could not be identified by the Mascot search engine and were assigned by manual interpretation (the complete spectral information is presented in Appendix A).

For all annotated peptides, manual interpretation revealed a good coverage of the peptide sequences by intense b- and y-fragment ions (Figure 3), whereas their glycated status, in the majority of cases, could be confirmed by the patterns of water and formaldehyde losses from the quasi-molecular ions, characteristic for glucose-derived Amadori compounds [40,41]. Additionally, the presence of a sugar moiety at the corresponding lysyl residues was confirmed by pyrylium Amadori-related fragment ions.

### 2.5. Linear Discriminant Analysis

First, the data were subjected to principle component analysis (PCA), and the relative influence of the extracted individual principle components (PCs) was addressed, leaving out only the first nine components, which are responsible for 95% of the sample variance (see the corresponding variable loadings listed in Appendix A). As can be seen from Figure 4, even the first two principal components provide decent class separation.

Alternatively, another procedure for the reduction of dimensionality, i.e., a variance inflation factor (VIF)-based factor filtering (following the workflow for generation of the LDA model presented in the Materials and Methods part), was applied to the original data. Therefore, diverse feature subsets could be generated depending on the order of the feature testing. In order to characterize possible non-collinear feature combinations, this algorithm was run 1000 times for the VIF cutoff values of 5 and 10. On the basis of the obtained results (Appendix A), the most abundant combinations of factors were selected for further training with LDA.

The results of the filtering procedure, based on the VIFs, are summarized in Appendix A, where individual variables are labeled as in Table 1. For the further processing, one of the most highly represented sets of variables with VIF = 10 and two sets, represented at 70% of all optimization runs with VIF = 5, were selected. Validation of the training and prediction accuracy was performed for the selected sets of variables in leave-one-out cross validation (LOOCV) and the sub-sampling LDA parameterizations modes (Figure 5). The resulting values for the accuracy, specificity, and sensitivity are summarized in Table 3.

## 3. Discussion

### 3.1. Experimental Setup

Protein glycation is one of the most universal markers of diabetes mellitus. It is recognized as an indirect, but adequate marker of blood glucose contents over a certain period of time [42]. Indeed, as blood sugar concentrations fluctuate in a wide range, depending on circadian rhythmus, food intake, medication, and other factors [43], the determination of actual blood glucose levels usually does not deliver clinically valuable information. In this context, HbA_1c_ is commonly used both in diagnostics of T2DM and long-term therapy control of the disease [42], whereas determination of short-term markers, like glycated albumin and total plasma fructosamine fraction [44,45], are still much less common in clinical practice. Thus, persisting hyperglycemia is typically confirmed only over the last three months prior to determination, whereas no or little information about the dynamics of glycation (and, hence, blood glucose levels) is available.

Therefore, the monitoring of several glycated proteins, varying in their half-life times, would be a promising alternative to the traditional approach for glycemic control. However, as the glycation rates at individual protein lysyl residues vary essentially [23], the total glycation levels of each protein would be less sensitive to sugar fluctuations than modification levels at individual residues. To address this concept in statistically representative cohorts, we analyzed the relative quantities of corresponding glycated species in the group (*n* = 20) of T2DM patients matched by age and gender to healthy controls. This cohort size was a good starting point for this pilot study, but due to a high number of potential marker peptides, we employed an α-level of 0.01 (i.e., lower, than usual) and implemented a family-wise error rate control procedure [46], thus ensuring that the samples sizes used in our study were sufficient to prove all related inferences. As for the PCA\LDA, dimensionality reduction techniques are well suited for the high parameters of samples ratio, and according to Sharma et al. [47] and references therein, our case (approximately 40 dimensions to 40 samples) is several orders of magnitude below typical thresholds, above which small sample size problems should be specifically addressed.

To access individual glycation sites, we applied an experimental setup relying on the bottom-up proteomic approach and LC-MS-based label-free quantification (Figure 2). As we have shown previously, the abundances of such Amadori peptides (i.e., the areas of corresponding peaks in appropriate XICs) can be used to quantify corresponding glycation sites in bovine serum albumin [25]. Moreover, testing of this approach in small cohorts by nano-scaled reversed-phase high performance liquid chromatography-electrospray ionization-tandem mass-spectrometry (nanoRP-HPLC-ESI-MS/MS) DDA experiments revealed the differential abundance of some glycation sites in T2DM patients and normoglycemic individuals, although the poor precision of the quantification restricted the reliability of the results’ interpretation [27].

Besides employing larger cohorts, this limitation can be generally overcome by using a reliable single-step chromatography system (i.e., skipping trapping procedure) as a part of a targeted or untargeted quantitative platform. Although the first approach, recently proposed by Spiller and co-workers [30], provides sufficient precision [29], it targets only a limited number of potential marker sites. Moreover, the selection of marker peptides, addressed in the multiple reaction monitoring (MRM) triple quadrupole (QqQ) method, proposed by the authors was based on our pilot study, which was performed in small cohorts consisting of only five probands [27]. This experimental setup could certainly result in missing many glycation sites with biomarker properties.

Therefore, here, we decided on an unbiased profiling approach based on high resolution mass spectrometry (HR-MS) coupled with on-line to microbore reversed-phase high performance liquid chromatography (RP-HPLC) with subsequent annotation of previously identified glycated peptides by the *m/z* of corresponding quasi-molecular ions. For this, we relied on the best available database of in vivo plasma glycation sites, built by Metz and co-workers on the basis of multi-dimensional in-depth plasma proteomics analysis [26]. Although in the absence of enrichment and pre-fractionation steps our straightforward approach (comprising boronic acid affinity chromatography (BAC) and RP-HPLC steps) yielded only 51 of the most abundant peptides, representing 46 modification sites among 2205 identified by nano-LC-MS/MS-based proteomics [26], their quantification was accurate and extremely precise. Indeed, our method provided a 14- and 7-fold improvement of intra- and inter-day precision for the peak area in comparison to the values, obtained by Frolov et al. (2014) [27], respectively (Appendix A). Thereby, inter-day precision values were even slightly better than those generated with QqQ-MS instrumentation [29] with an essentially higher number of considered peptides. Without any doubt, it positively affected the significance of differences between T2DM and control groups (Table 1, Figure 3 and Appendix A), with a simultaneous reduction of the analysis time and costs in comparison to the conventional LC-based proteomic approach.

In this context, the analytical workflow (Figure 2) was reduced to the combination of three principle steps—enrichment of glycated peptides with BAC, desalting of the samples by SPE, and subsequent reversed phase chromatography (RPC) separation [24,48]. In our recent work we proved that the the solid phase extraction (SPE) step is absolutely mandatory for the acceptable recovery of Amadori peptides from the reversed phase [29]. Therefore, it was ultimately included in the proposed workflow.

### 3.2. Biomarker Potential of Glycated Peptides

Although the size of the cohorts used in our previous study was insufficient for reliable conclusions, five individuals per comparison group was a good starting point to establish new biomarkers based on specific glycation sites in plasma proteins [27]. The increase of the cohort size in combination with an untargeted LC-MS approach resulted in the identification of a higher number of significantly up-regulated individual glycation sites with higher confidence (Table 1) in comparison to the published data, obtained with both spectral counting [49] and integration of specific signals in individual XICs [27]. Thus, our setup resulted in a higher biomarker potential of the plasma protein glycation sites considered in this context earlier [27,28,29] (Figure 6), and yielded new prospective Amadori peptides, whose biomarker behavior has not been reported yet.

As all these peptides belonged to proteins with varying half-life times, we succeeded in addressing the time scale of glycemic control (Table 2). Indeed, relatively long-living proteins (HSA and the immunoglobulin (Ig) kappa chain C region with τ_1/2_ of up to three weeks [39]) are exposed to plasma glucose for times much longer than those relevant for α-2-macroglobulin and complement C4-A protein (τ_1/2_ of 1 to 3 days [33,34]). Hence, the glycation sites representing the latter proteins would deliver valuable information about blood sugar levels over the days directly preceding the analysis. Such data would directly show if short-term fluctuations of plasma glucose levels do occur. Our results indicate a significantly lower degree of such fluctuations in the plasma of healthy individuals in comparison to T2DM patients (Table 1, Figure 6 and Appendix A). Thus, in the future, the corresponding tryptic peptides might be addressed as possible markers of impaired glucose tolerance.

### 3.3. Specific Set of Glycation Sites as an Integrated Biomarker of T2DM

As the proteins listed in Table 2 cover a large range of half-lives, consideration of the interference between representative glycation sites might provide a generalized T2DM marker, addressing a dynamic aspect of glycemic control. In this context, we tried to identify a set of multiple glycated sites in several marker proteins with various τ_1/2_. This set is expected to provide sensitivity sufficient for the prediction of T2DM and efficient glycemic control, i.e., provide separation of groups different from existing ones, e.g., glycated hemoglobin (Figure 7). However, a biomarker, based on multiple glycated proteins would give a higher degree in the flexibility of the glycemic control depth.

A basic T2DM marker, relying on two glycation sites, was proposed recently. Using a decision tree algorithm, Spiller and co-workers successfully combined quantitative information on a selected haptoglobin glycation site (K_141_) with a well-established T2DM marker, such as HbA_1c_ [30]. Unfortunately, due to its “step-by-step” decision tree design, this methodology is generally not applicable to the assessment of multiple factor contributions. Besides, when large cohorts are used, uncertainties, related to overfitting and multiple testing, need to be considered [49], i.e., a proper inspection of all resulting decision trees becomes critical for the accuracy of disease prediction. Indeed, as this calculation strategy does not assess factors simultaneously, the risk that the chosen set of classification steps and cut-off procedures “learns” some specific features of the training set is relatively high.

Therefore, to avoid a potential bias associated with the application of the decision tree design to multiple features, we decided on LDA as the main classification algorithm. The corresponding output variable represented a simple weighed sum of the analyzed features, which could successfully meet the requirements for the establishment of a time dimension in glycaemic control. Thereby, the procedures for selection of the model parameters aimed at the removal of highly-correlated sites (duplicates or linear combinations) in the feature space (VIF-based filtering) or for transformation of the feature space itself (PCA). We believe that the application of several complementary feature space transformation and randomization techniques (i.e., PCA, VIF-based filtering, random sub-sampling of the training set, and leave-one-out cross-validation) allowed us to exclude any experimental bias and ensure a high reliability of the prediction model set.

Generally, all model generation and validation procedures resulted in similar metric values (Table 3). It can be noted that the models generated with a VIF cutoff of 5 demonstrated a clearly better performance (sensitivity, accuracy, and specificity of 80%, 90%, and 100%, respectively), in comparison to the models with a VIF cutoff of 10, containing higher numbers of features. Performance of the PCA-based model was somewhere in the middle, possibly due to the averaging of the specifics of individual features upon calculation of the principle component (PC) variable values. It is important to stress that the diagnostic results presented in Table 3 are conservative, i.e., bias-free [49]. Also, as the sub-sampling training sets discard the information about feature cross-correlations, they might be less informative than ones used for leave-one-out cross-validation, thus providing less accurate predictions.

Also, in the future, for the collection of blood samples, it is necessary to keep in mind not only therapy, but also the diet of patients. Thus, the influence of dietary polyphenols on blood glucose at different levels may also help control and prevent diabetes complication via a decrease of hyperglycemia and an improvement of acute insulin secretion and insulin sensitivity [50].

## 4. Materials and Methods

### 4.1. Reagents

Unless stated otherwise, materials were obtained from the following manufacturers. AMRESCO LLC (Fountain Parkway, Solon, OH, USA): ammonium persulfate (ACS grade), glycine (biotechnology grade), *N,N*ʹ-methylene-bis-acrylamide (ultra-pure grade), tris(hydroxymethyl)aminomethan (tris, ultra-pure grade); Aptec Diagnostics nv (Saint-Petersburg, Russia): Albumin kit; Becton Dickinson (Moscow, Russia): BD Vacutainer^®^ Plus Plastic K2EDTA tubes; Carl Roth GmbH & Co (Karlsruhe, Germany): acetonitrile (≥99.95%, LC-MS grade), formic acid (≥98%, p.a., ACS), ethanol (≥99.8%), methanol (HPLC Ultra Gradient Grade), sodium dodecyl sulfate (SDS) (>99%), tris-(2-carboxyethyl)-phosphine hydrochloride (TCEP, ≥98%); Component-Reactiv (Moscow, Russia): phosphoric acid (analytical quality); PanReac AppliChem (Darmstadt, Germany): acrylamide (2K Standard Grade), glycerol (ACS grade); QIAGEN GmbH (Hilden, Germany): 1 mL polypropylene columns; Reachem (Moscow, Russia): Hydrochloric acid (analytical quality); Roche Diagnostics (Saint-Petersburg, Russia): Precinorm U plus Control, Precipath U plus Control; SERVA Electrophoresis GmbH (Heidelberg, Germany): Coomassi Brilliant Blue G-250, 2-mercaptoethanol (research grade), trypsin NB (sequencing grade, modified from porcine pancreas); Thermo Fisher Scientific (Waltham, MA, USA): PageRuler™ Unstained Protein Ladder #26614, PageRuler™ Prestained Protein Ladder #26616; Vekton (Saint-Petersburg, Russia): ammonia hydroxide (ACS grade); Waters GmbH (Eschborn, Germany): Oasis HLB cartridges (10 mg, 30 µm particle size). All other chemicals were purchased from Sigma-Aldrich Chemie GmbH (Taufkirchen, Germany). Water was purified in house (resistance >16 mΩ/cm) on a water conditioning and purification system “UVOI-MF-1-NA(18)-N” (Mediana-Filtr, Moscow, Russia).

### 4.2. Setup of Experimental Cohorts

The T2DM patient (*n* = 20) and normoglycemic control (*n* = 18) groups comprised non-smoking female volunteers aged 45 to 75 years (63.4 ± 7.9 and 60.7 ± 4.7, respectively), who were not receiving hormone replacement therapy and had no clinically manifested diabetes complications (Table 4 and Table 5). The control individuals’ HbA_1C_ levels did not exceed 6.5%, and did not have diagnosed diabetes and anti-hyperglycemic therapy in their medical history. All participants provided written informed consent. The study was approved 02-03-2015 by the Local Ethical Committee of the Federal Almazov North-West Medical Research Centre, Saint-Petersburg, Russian Federation, and was performed in agreement with the Declaration of Helsinki.

### 4.3. Blood Sampling and Plasma Isolation

The blood samples (approximately 10 mL each) were collected in polypropylene tubes coated with etylenediaminetetraacetic acid (Becton Dickinson, Franklin Lakes, NJ, USA). Plasma was separated by centrifugation (1200× *g*, 15 min, 4 °C) and transferred to 1.5 mL polypropylene tubes. The total plasma protein contents were determined by Bradford assay in a 96-well microtiter plate format as described by Greifenhagen and co-workers [51]. The precision of protein determination was verified by SDS-PAGE according to an established protocol [11]. Average densities across individual lanes (expressed in arbitrary units) were determined by a ChemiDoc XRS imaging system controlled by Quantity One^®^ 1-D analysis software (Bio-Rad Laboratories Ltd., Moscow, Russia). Thereby, for inter-gel normalization, the first and the last plasma protein samples loaded on each gel were replicated in the previous and following gels, respectively (Appendix A). For the calculation of RSDs, the densities of individual lines were normalized to the gel average value. Individual plasma samples were split into aliquots of 20 µL, and stored at −80 °C. Alternatively, 30 µL of each T2DM sample were combined to obtain a representative pool of diabetic material. The plasma contents of human serum albumin were determined calorimetrically at 628 nm after a color reaction with bromocresol purple (Clinical Chemistry Analyzer CA90, Furuno Electric Co. LTD, Nishinomiya, Japan) using an Albumin kit (Aptec Diagnostics nv) and the control serum kits, Precinorm U and Precipath U (Roche Diagnostics). The plasma HbA_1c_ level was measured using the commercial kit, BioRad (Hercules, CA, USA), on a hemoglobin analyzer BioRad D-10 (Bio-Rad Laboratories Inc, Hercules, CA, USA).

### 4.4. Tryptic Digestion

The plasma proteins were digested according to Frolov and co-workers [27] with slight modifications. Briefly, aliquots of plasma containing 150 μg of protein were diluted with 100 mmol/L ammonium bicarbonate buffer (pH 8.0), complemented with 10 µL of SDS (0.5% in water, *w*/*v*) and 10 µL of 50 mmol/L TCEP in 100 mmol/L ammonium bicarbonate buffer, and incubated for 30 min at 37 °C under continuous shaking (450 rpm). Afterwards, the samples were cooled to room temperature (RT), and 11 µL of 100 mmol/L iodoacetamide in 100 mmol/L ammonium bicarbonate buffer were added, and alkylation of free sulfhydryls was performed during 15 min in darkness at RT. After completion of the incubation, the proteins were sequentially digested at 37 °C with trypsin (25 μg/mL in 100 mmol/L ammonium bicarbonate buffer) taken in the 1:20 and 1:40 (*w*/*w*) enzyme-protein ratio for 5 and 12 h, respectively, under continuous shaking (450 rpm).

The completeness of the digest was verified by SDS-PAGE as described by Schmidt and co-workers [52] with modifications. Briefly, aliquots of digested samples containing 5 μg of protein were diluted with sample buffer (65.8 mmol/L Tris-HCl, pH 6.8, 20% (*v*/*v*) glycerol, 2% SDS, 10% (*v*/*v*) β-mercaptoethanol, 0.05% (*v*/*v*) bromophenol blue) at least 2-fold and heated at 95 °C for 5 min. Afterwards, the samples were separated on a polyacrylamide gel (T = 12.00%, C = 2.65%), and stained with colloidal Coomassie Brilliant Blue G 250 dye. The digests were frozen and stored at −80 °C before further analysis.

### 4.5. Boronic Acid Affinity Chromatography

Enrichment of glycated peptides was performed according to Soboleva and co-workers [28] with slight modifications. In detail, the pH of tryptic digests was adjusted to 8.0 with 25% (*v*/*v*) ammonia hydroxide using indicator paper (Lachema, Brno, Czech Republic), before 400 μL of ice cold (4 °C) loading buffer (250 mmol/L ammonium acetate, 50 mmol/L magnesium acetate, pH 8.1) were added. The samples were loaded on 1 mL polypropylene gravity flow columns packed with m-aminophenylboronic acid (mAPBA) agarose, and unbound peptides were washed out with 12 mL of ice-cold (4 °C) loading buffer. Afterwards, glycated peptides were sequentially eluted with 0.1 and 0.2 mol/L warm (37 °C) acetic acid (8 and 2 mL, respectively). The eluates were combined, and loaded on Oasis HLB SPE cartridges, installed on the VacElut 12 Manifold (Agilent Technologies, Moscow, Russia), pre-conditioned with 1 mL of methanol, and pre-equilibrated with 2 mL of 0.1% (*v*/*v*) aqueous (aq.) formic acid. After a wash with 2 mL of 0.1% (*v*/*v*) formic acid, peptides were eluted in a step gradient of 40%, 60%, and 80% acetonitrile (0.33 mL each), as described by Spiller et al. [29]. The SPE-eluates were combined, dried under vacuum by a CentriVap Vacuum Concentrator (Labconco, Kansas City, MO, USA), and stored at −20 °C for further analysis.

### 4.6. LC-MS Analysis

The dried eluates were reconstituted in 40 µL of 3% (*v*/*v*) acetonitrile in aq. 0.1% (% *v*/*v*) formic acid, and 8 µL of the obtained solutions were loaded on a ZORBAX SB column (C18, ID 0.3 mm, length 150 mm, particle size 3.5 μm, Agilent Technologies, Moscow, Russia) using an Agilent 1200 Compact liquid chromatograph equipped with an Agilent 1200 Infinity autosampler and Agilent 1260 Infinity capillary pump (Agilent Technologies, Moscow, Russia). The eluents, A and B, were 4% and 90% acetonitrile, respectively, both containing 0.1% (*v*/*v*) formic acid. After a 5-min isocratic step (0% eluent B), glycated peptides were eluted at the flow rate of 5 µL/min at 25 °C in sequential linear gradients to 45% and to 100% eluent B in 30 and 2 min, respectively. The column effluents were introduced on-line in an Agilent 6538 Ultra High Definition Accurate-Mass Q-TOF quadrupole-time of flight (QqTOF) mass spectrometer via a dual electrospray ionization (ESI) source (Agilent Technologies, Moscow, Russia). The instrument was operated in the positive ion mode under the settings summarized in Appendix A and controlled by MassHunter Workstation software (Agilent Technologies, Moscow, Russia). Analyte annotation and label free quantification relied on TOF-MS scans acquired in the mass range of 400–2000 *m/z*. Thereby, a pooled enriched tryptic digest, obtained from T2DM patients, was used as an external QC, injected after each eighth sample. Prospective Amadori-modified peptides were annotated by t_R_s and exact *m/z* values (mass accuracy better than 3 ppm). Relative abundances of individual prospectively glycated peptides were calculated by integration of characteristic extracted ion chromatograms (XICs, *m/z* ± 0.02) built for the annotated *m/z* values at specific t_R_s (quantitative analysis tool of the MassHunter Workstation software).

### 4.7. MS/MS Analysis

The sequences and glycation status of differentially abundant glycated peptides were confirmed by MS/MS using a combination of DDA and targeted MS/MS experiments using an LTQ-Orbitrap Velos Pro mass spectrometer. For this, a digest of pooled T2DM plasma (1.5 µg) was loaded on the Acclaim® PepMap100 pre-column (C18-phase, ID 75 μm, length 2 cm, particle size 3 µm), and separated on the EASY-Spray ES803 C18 column (500 × 0.075 mm, 2 µm particle size, 40 °C) using an EASY-nLC 1000 nano liquid chromatography system controlled by the Xcalibur 2.1.0 software. Eluents A and B were water and acetonitrile, respectively, both containing 0.1% (*v*/*v*) formic acid. The analytes were eluted at the flow rate of 0.3 µL/min as follows: 5% B (0–15 min), 5% to 40% B (15–195 min), 40% to 80% B (195–197 min). The column effluent was transferred on-line to MS (operated in positive ion mode and controlled by Xcalibur 2.1.0 software) via an EASY-spray ion source. DDA analyses relied on survey Orbitrap-MS scans followed by dependent linear ion trap (LIT) ones (collision induced dissociation (CID) fragmentation, He, 35% normalized collision energy) acquired for the 20 most intense signals with charge states of two or more. For targeted MS/MS, 10 LIT MS2 sub-experiments per liquid chromatography-tandem mass-spectrometry (LC-MS/MS) run were acquired for the two most intense quasi-molecular ions of each peptide, not fragmented in DDA experiments. The MS settings are summarized in Appendix A. Tandem mass spectra were searched against a FASTA file containing sequences of the proteins annotated by QqTOF-MS using the Mascot search engine within Proteome Discoverer 1.4 software (Thermo-Fisher Scientific, Bremen, Germany) with the following settings: Peptide tolerance—7 ppm, MS/MS match tolerance—0.8 Da, 3 missed cleavage sites per peptide. Glycated peptides were annotated by their t_R_s, *m*/*z* values, and isotopic patterns. The results were filtered with consideration of peptide confidence (medium), rank (one), false positive rate (0.05), and post-translational modifications—carbamidomethylation (C), oxidation (M), and glycation (K).

### 4.8. Statistical Analysis

The statistical significance of differences in the relative abundances of specific glycated peptides detected in the plasma protein digests obtained from T2DM patients and healthy individuals was determined by the Mann-Whitney *U* test [53]. Holm-Bonferroni correction was applied to adjust significance in terms of multiple hypothesis tests [46]. Since the intensities observed for the signals representing individual charge states of the same peptides demonstrated high correlation, these values were averaged for further processing. To distinguish the T2DM patients from the age-matched controls without diabetes, a linear discriminant analysis (LDA) approach was applied as described by Venables and Ripley [54]. All LDA-related procedures were performed in R (MASS package); all other calculations were performed in MS Excel with RealStatisticsaddon (http://www.real-statistics.com).

### 4.9. Generation of the LDA Model

Because the initial correlation analysis of the feature space showed a high degree of multicollinearity between factors (see Appendix A for a factor cross-correlation plot), we employed two strategies for feature space optimization prior to LDA calculation: (i) PCA-based transformation of the feature space, and (ii) generation of an orthogonal set of original features by the removal of factors with high variance inflation factor (VIF) values [55]. The latter can be calculated for any variable in the set (1) and is a numeric measure of to which extent the variable in question can be predicted by the linear combination of other variables, with the most commonly used critical VIF values being 5 and 10, corresponding to the multiple correlation of 90% and 95% between the tested variables and the rest variables in the set.
(1)$$VIF_{i}=\frac{1}{1-R_{i}^{2}},$$
where: *VIFi*—variance inflation factor for variable *i*; $R_{i}^{2}$—coefficient of determination, i.e., the proportion of the variance in the variable, *I*, which is predictable based on another independent variable

The application of PCA was straightforward and only the PCs responsible for 95% of the original sample variance were selected. The corresponding values were estimated for each sample, and the resulting reduced feature set was subjected to further processing with LDA. The VIF-based filtering relied on an iterative procedure, employing a random subset of two features (a test sub-set), and defined in the beginning of each calculation run. Afterwards, further features were randomly selected from the original set and sequentially included in the test subset in each new analysis run. Then, individual features with the VIF values above a selected cutoff (set as described above [43]) were iteratively removed from the test subset in the order of decreasing values. This procedure was further applied in this way to test all initial features.

### 4.10. Validation of the LDA Model

To assess the model performance, two complementary approaches were employed. In terms of the first one, accuracy, sensitivity, and specificity were determined for the established set of diagnostic peptides. To avoid the overfitting problem [56], parameterization of the LDA model was performed in silico with a set of 400 calculated sample patterns (200 for each cohort), obtained by the sub-sampling feature distributions of original samples as described by Politis and co-authors [57]. Thereby, all selected training samples shared less than 25% of the factor values, obtained with any of the original (verification) vectors. The second approach relied on the so-called leave-one-out cross-validation procedure [58]. It employed a set of iterations, where at each run one sample from the original sample set was taken for verification, whereas the other samples were used for parameterization of the LDA model. The performance characteristics (i.e., accuracy, sensitivity, and specificity) were calculated from the set of individual iteration outcomes as described elsewhere [59,60]. For the PCA-generated feature subset, only the leave-one-out cross-validation procedure was performed.

## 5. Conclusions

T2DM is one of the most widely spread metabolic disorders. Typically, the first stages of the disease are slow and are not accompanied by any clinically manifested symptoms. Because of this reason, T2DM is most often discovered at the step of complications, which makes therapy less efficient and more expensive. Unfortunately, HbA_1c_—a recognized T2DM marker—delivers information about changes in glycaemic status over three months, and, hence, is insensitive to short-term glucose excursions preceding the disease. Thus, control of glycemic status over various, especially short, periods of time might increase the rates of early T2DM discovery. In this context, our approach might bring the desired “time dimension” in glycemic control. Indeed, the integrated biomarker, proposed here, not only covers three weeks before blood sampling, but also indicates a continuous character of the glycation process throughout this period. Secondly, although our integrated biomarker relies on multiple glycation sites, all required information is acquired in one experiment, which is advantageous in comparison to the approaches employing several tests. Finally, the proposed marker has potential for further optimization. Thus, it can be “tuned” for shorter times by excluding relatively long-living proteins. On the the other hand, implementation of immunoaffinity depletion might essentially increase the pattern of marker peptides, and, hence, the selection of marker proteins and the reliability of the marker.

## Figures and Tables

**Figure 1 ijms-20-02329-f001:**
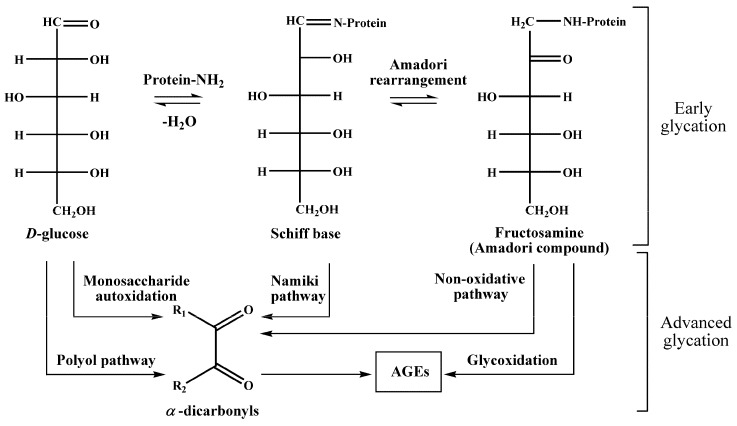
Early and advanced glycation in human blood plasma. The major pathways of advanced glycation end product (AGE) formation: monosaccharide autoxidation (Wolff-pathway), autoxidation of aldimins (Namiki-pathway), polyol pathway, autoxidation of Amadori products (Hodge-pathway), and non-oxidative pathway.

**Figure 2 ijms-20-02329-f002:**
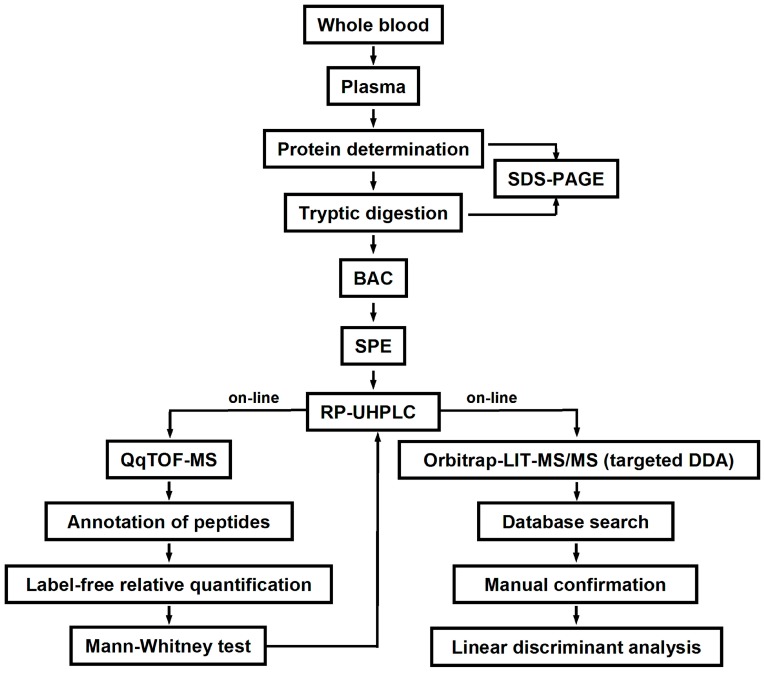
Overview of the experimental workflow.

**Figure 3 ijms-20-02329-f003:**
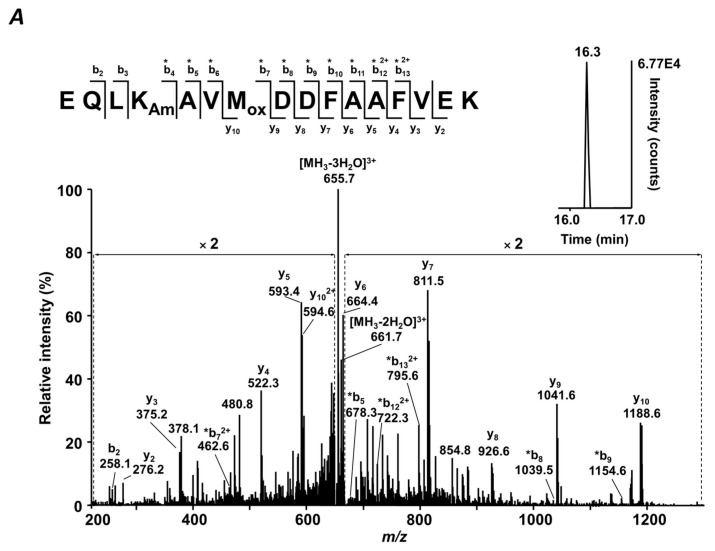
Tandem mass spectra acquired at *m/z* 673.66 (**A**) and 463.56 (**B**) corresponding to the [M + 3H]^3+^ ions of the peptides, EQLK_Am_AVM_ox_DDFAAFVEK and FK_Am_DLGEENFK (both human serum albumin). The spectra were acquired with the LTQ-Orbitrap Velos Pro mass spectrometer operated in the positive ion mode as a part of targeted RP-HPLC-Orbitrap-LIT-MS/MS DDA workflow. The inserts represent extracted ion chromatograms (*m/z* ± 0.02) acquired for the *m/z* 673.66 (**A**) and 463.56 (**B**). K_Am_ and M_Ox_ denote glycated lysyl and oxidized methionyl residues, respectively.

**Figure 4 ijms-20-02329-f004:**
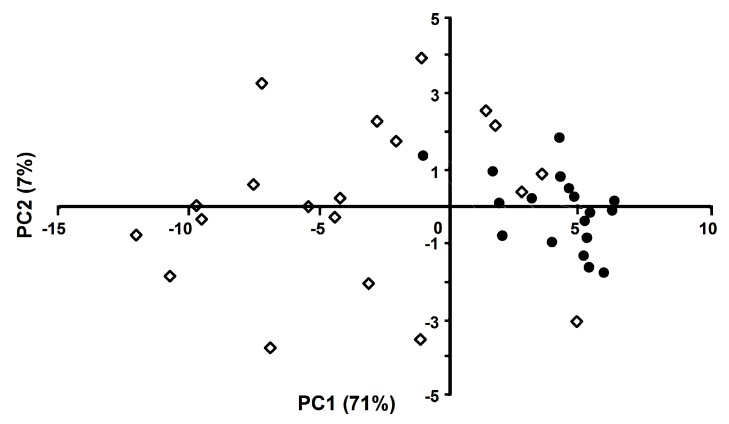
Results of the principle component analysis (PCA) performed for T2DM patient (opened diamonds) and normoglycemic (closed circles) groups. The analysis was based on all 42 differentially glycated peptides and indicated clear separation between the groups.

**Figure 5 ijms-20-02329-f005:**
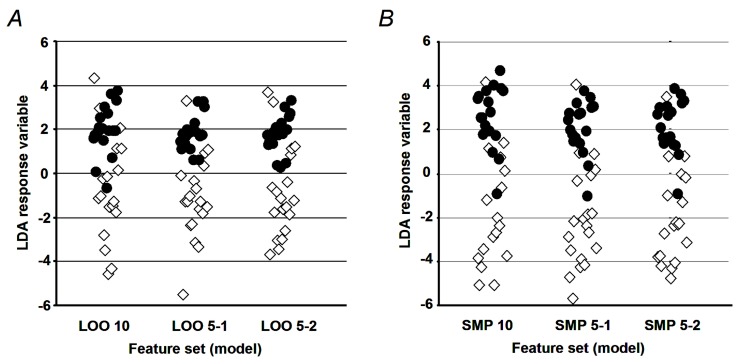
Values of the linear discriminant analysis (LDA) response variable plotted for T2DM (opened diamonds) and control (closed circles) individuals, calculated using selected predictive variable sets (as summarized in Table 3). (**A**) Estimated via leave-one-out cross-validation; (**B**) estimated from original samples, with LDA trained on generated samples.

**Figure 6 ijms-20-02329-f006:**
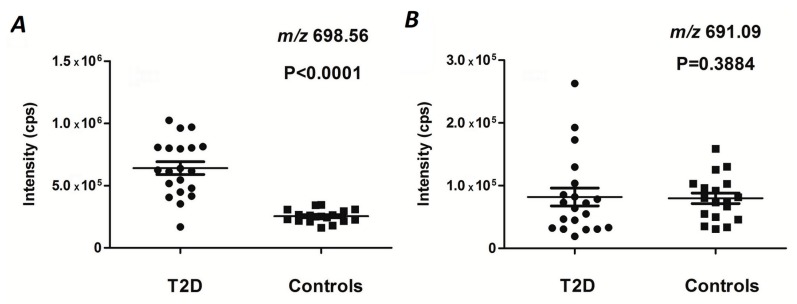
Peak areas of the signals at 38.7 min (**A**), and 48.9 min (**B**), integrated in the extracted ion chromatograms (XICs) of *m/z* 698.56 ± 0.02 (**A**) and 691.09 ± 0.02 (**B**), corresponding to the [M + 4H]^4+^ ions of glycated peptides LVNEVTEFAK_Am_TCCAMVADESAENCCAMDK (human serum albumin, **A**) and QNCCAMELFEQLGEYK_Am_FQNALLVR (human serum albumin, **B**), respectively. The data were acquired in the RP-HPLC-QqTOF-MS experiments performed with boronic acid affinity chromatography (BAC) enriched tryptic digests obtained from individual T2DM and control plasma samples. Statistical significance was estimated by the Mann-Whitney *U*-test, and *p* values below 0.05 were considered to be significant. K_Am_ and C_CAM_ denote glycated lysyl and carbamidomethylatedcysteinyl residues, respectively.

**Figure 7 ijms-20-02329-f007:**
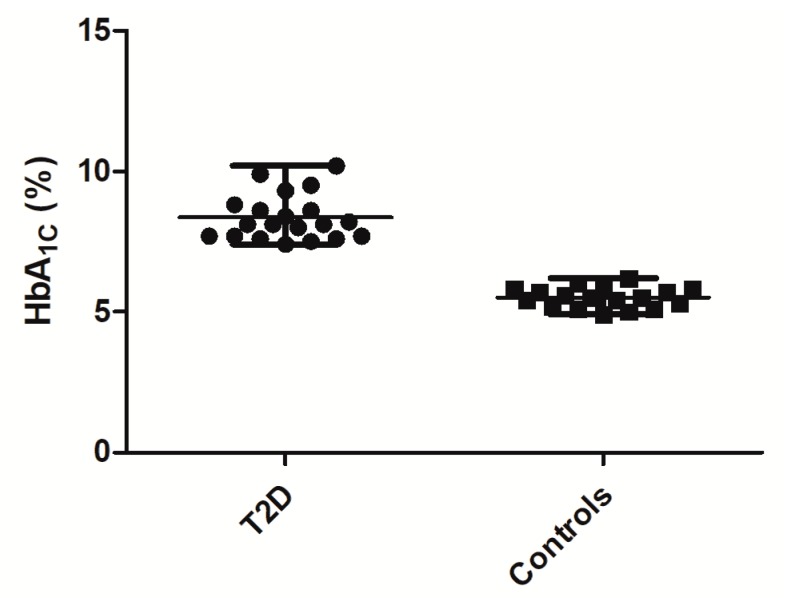
The level of HbA_1c_ in plasma samples obtained from patients with T2DM and healthy persons. Statistical significance was estimated by the Mann-Whitney *U*-test, *p* < 0.0001.

**Table 1 ijms-20-02329-t001:** Individual glycated peptides differentially abundant in tryptic digests obtained from the T2DM patients and individuals without diabetes.

#	Sequence	Glycation Site	*t*_R_ (min)	*m/z_obs_*	*z*	Error (ppm)	U-Statistic (*z*)	*p* Value (*z*)	Ac	Sp	Sn	LDA Variable
Human serum albumin												
1	VTK*C*C*TESLVNR	K_475_	14.5	543.5926	3	0.2	45, (3)	<0.0001, (3)	81.6	77.7	85	V24
2	RYK*AAFTEC*C*QAADK	K_162_	22.5	495.9759/660.9649	4/3	0.4/0.2	12, (4)	<0.0001, (4)	97.4	100	95	V14
3	LK*EC*C*EKPLLEK	K_276_	24.4	427.9688/570.2889/854.9281	4/3/2	0.9/0.5/1.0	19, (3)	<0.0001, (3)	86.8	83.3	90	V20
4	C*ASLQK*FGER	K_205_	25.4	453.2191/679.3240	3/2	0.2/1.5	14, (3)	<0.0001, (3)	92.1	94.4	90	V22
5	LAK*TYETTLEK	K_351_	25.5	486.9246/729.8828	3/2	0.2/0.0	32, (3)	<0.0001, (3)	84.2	77.7	90	V15
6	YK*AAFTEC*C*QAADK	K_162_	26.2	608.9315/912.8924	3/2	0.0/0.8	21, (3)	<0.0001, (3)	86.8	100	75	V16
7	ETYGEMADC*C*AK*QEPER	K_93_	28.0	745.9667	3	0.0	43, (3)	<0.0001, (3)	84.2	94.4	75	V27
8	YIC*ENQDSISSK*LK	K_274_	29.9	616.2968	3	0.2	44, (3)	<0.0001, (3)	86.8	75	100	V25
9	FK*DLGEENFK	K_12_	31.0	463.5581/694.8330	3/2	0.2/0.1	44, (3)	<0.0001, (3)	81.6	77.7	85	V17
10	TYETTLEK*C*C*AAADPHEC*YAK	K_359_	31.7	670.7864/894.0452	4/3	0.1/0.7	45, (4)	<0.0001, (4)	79.0	66.6	90	V28
11	K*YLYEIAR	K_137_	31.9	406.5524/609.3246	3/2	0.0/0.3	22, (3)	<0.0001, (3)	86.8	88.8	85	V18
12	K*QTALVELVK	K_525_	33.1	430.9227/645.8801	3/2	0.5/0.3	46, (3)	<0.0001, (3)	84.2	88.8	80	V19
13	K*VPQVSTPTLVEVSR	K_414_	35.5	601.3355/901.4988	3/2	0.0/0.7	18, (3)	<0.0001, (3)	89.5	94.4	85	V11
14	ADLAK*YIC*ENQDSISSK	K_262_	36.0	701.9975	3	0.7	49, (3)	0.0001, (3)	86.8	100	75	V1
15	LVTDLTK*VHTEC*C*HGDLLEC*ADDR	K_240_	36.1	604.6759/755.5923/1007.1200	5/4/3	1.2/0.4/0.3	43, (5)	<0.0001, (5)	84.2	70	100	V23
16	TC*VADESAENC*DK*SLHTLFGDK	K_64_	37.4	665.5443/887.0572	4/3	1.4/2.1	34, (4)	<0.0001, (4)	84.2	66.6	100	V2
17	K*LVAASQAALGL	K_574_	37.5	652.3774	2	0.2	56, (2)	0.0003, (2)	79.0	66.6	90	V12
18	AAFTEC*C*QAADK*AAC*LLPK	K_174_	38.5	763.0193	3	0.7	30, (3)	<0.0001, (3)	84.2	66.6	100	V3
19	LVNEVTEFAK*TC*VADESAENC*DK	K_51_	38.7	698.5634/931.0817	4/3	0.4/0.2	17, (4)	<0.0001, (4)	84.2	94.4	75	V4
20	AAC*LLPK*LDELR	K_181_	40.9	520.9491/780.9191	3/2	0.8/0.1	40, (3)	<0.0001, (3)	86.8	88.8	85	V21
21	SLHTLFGDK*LC*TVATLR	K_73_	42.5	524.2790/698.7018	4/3	1.0/0.3	41, (3)	<0.0001, (3)	81.6	83.3	80	V13
22	EFNAETFTFHADIC*TLSEK*ER	K_519_	42.9	677.5613/903.0777	4/3	0.7/0.8	67, (4)	0.0010, (4)	73.7	55.5	90	V26
23	TC*VADESAENC*DK*SLHTLFGDKLC*TVATLR	K_64_	43.1	894.1641	4	5.0	51, (4)	0.0001, (4)	81.6	72.2	90	V29
24	EQLK*AVM*DDFAAFVEK	K_545_	43.9	673.6598	3	0.3	68, (3)	0.0011, (3)	76.3	88.8	65	V7
25	VFDEFK*PLVEEPQNLIK	K_378_	45.8	736.3884	3	0.4	80, (3)	0.0036, (3)	76.3	72.2	80	V5
26	AEFAEVSK*LVTDLTK	K_233_	48.5	604.9884	3	0.8	79, (3)	0.0033, (3)	79.0	88.8	70	V6
27	RHPYFYAPELLFFAK*	K_159_	50.1	516.0182	4	0.4	86, (4)	0.0063, (4)	73.7	44.4	100	V30
Alpha-2-macroglobulin												
28	LVDGK*GVPIPNK	K_375_	28.5	466.9332	3	0.2	33, (3)	<0.0001, (3)	86.8	94.4	80	V32
29	ALLAYAFALAGNQDK*R	K_1162_	46.2	628.6674	3	0.0	77, (3)	0.0027, (3)	76.3	66.6	85	V10
Apolipoprotein A1												
30	LAEYHAK*ATEHLSTLSEK	K_219_	28.5	548.2794	4	0.5	80, (4)	0.0036, (4)	76.3	94.4	60	V39
Ceruloplasmin precursor												
31	VTFHNK*GAYPLSIEPIGVR	K_468_	39.9	565.8053	4	0.0	57, (4)	<0.0001, (4)	84.2	70	100	V40
Complement C4-A												
32	RC*SVFYGAPSK*SR	K_1575_	39.0	838.9029	2	5.0	27, (2)	<0.0001, (2)	84.2	72.2	95	V42
FLJ00385 protein												
33	VSNK*ALPAPIEK	K_319_	26.6	476.9368/714.9006	3/2	0.0/1.0	30, (3)	<0.0001, (3)	86.8	75	100	V38
Igkappachain C region												
34	VQWK*VDNALQSGNSQESVTEQDSK	K_41_	34.3	947.1123	3	0.6	51, (3)	0.0002, (3)	81.6	94.4	70	V8
35	DSTYSLSSTLTLSK*ADYEK	K_75_	38.7	757.6970	3	0.5	35, (3)	<0.0001, (3)	86.8	88.8	85	V9
Serotransferrin												
36	GDVAFVK*HQTVPQNTGGK	K_553_	25.7	512.0137	4	0.6	37, (4)	<0.0001, (4)	86.8	94.4	80	V31
37	DDTVC*LAK*LHDR	K_659_	29.0	401.9458/535.5914	4/3	0.5/0.2	25, (4)	<0.0001, (4)	84.2	94.4	75	V36
38	KPVDEYK*DC*HLAQVPSHTVVAR	K_258_	30.2	543.0768/678.5909/904.4496	5/4/3	2.4/2.2/4.8	34, (5)	<0.0001, (5)	86.8	88.8	85	V37
39	SK*EFQLFSSPHGK	K_299_	34.3	414.2098/551.9418	4/3	2.1/1.4	42, (3)	<0.0001, (3)	81.6	77.7	85	V34
40	DLLFK*DSAHGFLK	K_315_	40.3	413.9693/551.6227	4/3	0.2/0.5	47, (4)	0.0001, (4)	81.6	83.3	80	V35
41	DGAGDVAFVK*HSTIFENLANK	K_225_	41.6	599.5509/799.0649	4/3	0.8/0.5	53, (3)	0.0002, (3)	84.2	94.4	75	V33
Vitamin D-binding protein precursor												
42	TSALSAK*SC*ESNSPFPVHPGTAEC*C*TK	K_94_	32.3	772.0957	4	1.3	39, (4)	<0.0001, (4)	86.8	83.3	90	V41

Individual tryptic digests obtained from T2DM patients (*n* = 20) and corresponding controls (*n* = 18) were analyzed by capillary RP-HPLC-ESI-QqTOF-MS. The relative quantification relied on a label-free strategy using a digest prepared from T2DM pooled plasma as a quality control (injected after each 8th sample). Statistical analysis relied on the Mann-Whitney test. C*, M*, and K* denote carbamidomethylated cysteine, methionine sulfoxide, and fructosamine lysine residues, respectively. Ac, Sp, and Sn denote the mass accuracy, accuracy, specificity, and sensitivity, respectively (calculated in MS Excel with RealStatisticsaddon, http://www.real-statistics.com).

**Table 2 ijms-20-02329-t002:** The half-life times of plasma proteins, containing specific glycation sites differentially abundant in T2DM patients and normoglycemic individuals.

Glycated Protein	Glycation Site	Representativemarker Peptide (#)	*τ_1/2_* (Days)	Ref.
α-2-macroglobulin	K_1162_	ALLAYAFALAGNQDK*R (29)	1–2	[33]
Complement C4-A	K_1575_	RC*SVFYGAPSK*SR (32)	2–3	[34]
Vitamin D-binding protein precursor	K_94_	TSALSAK*SC*ESNSPFPVHPGTAEC*C*TK (42)	2.5–3	[35]
Ceruloplasmin	K_468_	VTFHNK*GAYPLSIEPIGVR (31)	5.5	[36]
Apolipoprotein A1	K_219_	LAEYHAK*ATEHLSTLSEK (30)	5–6	[37]
Serotransferrin	K_299_	SK*EFQLFSSPHGK (39)	8–10	[38]
Igkappachain C region	K_75_	DSTYSLSSTLTLSK*ADYEK (35)	19–21	[39]
Human serumalbumin	K_159_	RHPYFYAPELLFFAK* (27)	19–21	[39]
FLJ00385 protein	K_319_	VSNK*ALPAPIEK (33)	no data	-

**Table 3 ijms-20-02329-t003:** Summary of the predictor performance validation under distinct training and performance estimation conditions.

Model Type	Training\Prediction Mode						Variable Set
	LOO Cross Validated			Sub-Sample Trained			
	Ac	Sp	Sn	Ac	Sp	Sn	
VIF10	81.58	88.89	75	84.21	94.44	75	V6 V7 V9 V10 V12 V28 V29 V30 V32 V33 V34 V39 V40 V41 V42
VIF5_1	92.11	100	85	86.84	94.44	80	V6 V10 V12 V28 V29 V30 V33 V34 V39 V41 V42
VIF5_2	89.47	100	80	89.47	94.44	85	V7 V10 V12 V28 V29 V30 V32 V33 V34 V39 V41 V42
PCA	84.2	100	70	-	-	-	PC1 PC2 PC3 PC4 PC5 PC6 PC7 PC8 PC9

Ac, accuracy; Sp, specificity; Sn, sensitivity.

**Table 4 ijms-20-02329-t004:** Physical and biochemical parameters of the T2DM patients, as well as corresponding diabetes-related and non-related therapy.

Participants	Height (cm)	Weight (kg)	Body Mass Index	Age (Years)	Disease Duration (Years)	Therapy		HbA_1C_ (%)	Content of Albumin (μg/μL) in Blood Serum	Complications of Type 2 Diabetes Mellitus
						Diabetes-Specific Therapy	Diabetes-Nonspecific Therapy			
1	157	60.3	24.5	55	13	Gliclazide, Metformin	Verapamil, Enalapril, Hydrochlorothiazide, Atorvastatin	8.6	44.4	Polyneuropathy, nephropathy
2	165	97	35.6	50	14	Metformin, Vildagliptin	Valsartan, Verapamil, Rosuvastatin, Fenofibrate	9.5	35.2	Neuropathy, nephropathy, retinopathy
3	162	89.5	34.1	64	7	Metformin, Vildagliptin	Atorvastatin, Perindopril, Indapamide	8.0	44	Neuropathy
4	175	98.5	32.2	56	8	Metformin, Gliclazide	Losartan, Levothyroxine, Acetylsalicylicacid, Magnesiumhydroxide	8.6	48	Neuropathy
5	155	80.0	33.3	68	10	Glimepiride, Metformin, Vildagliptin	Perindopril, Bisoprolol, Amlodipine, Atorvastatin, Acetylsalicylicacid	8.8	34.6	Neuropathy, retinopathy, nephropathy
6	164	105	39.0	65	14	Metformin, Vildagliptin, Glimepiride	Acetylsalicylicacid, Losartan, Bisoprolol, Amlodipine, Moxonidine, Furosemide, Atorvastatin, Fenofibrate, Doxazosin	7.5	49	Neuropathy, retinopathy, nephropathy
7	163	72	27.0	57	1	Liraglutide	Rosuvastatin	8.1	48	-
8	166	76	27.6	72	3	Metformin	Amlodipine, Acetylsalicylic acid	7.4	49.3	Neuropathy
9	157	70	28.4	65	5	Metformin	-	7.6	41.4	Neuropathy, retinopathy
10	158	105.0	42	72	10	Metformin, Glibenclamide	Metoprolol, Digoxin, Atorvastatin, Famotidine, Dabigatran, Torasemide, Azilsartan	8.1	47	Neuropathy
11	166	69	25.0	65	10	Metformin, Liraglutide	Moxonidine, Rosuvastatin	7.7	49.4	Neuropathy
12	156	72	29.5	58	5	Metformin, Glimepiride, Vildagliptin	Losartan	8.4	46.7	Neuropathy
13	156	67	27.5	72	9	Metformin	Levothyroxine, Perindopril, Indapamide, Metoprolol, Acetylsalicylicacid, Magnesiumhydroxide, Fenofibrate, Rosuvastatin	7.7	48.4	Neuropathy
14	160	76	29.7	57	10	Metformin	Hydrochlorothiazide, Eprosartan, Acetylsalicylicacid, Magnesiumhydroxide	8.1	48.3	Neuropathy
15	167	89	31.9	67	10	Metformin, Sitagliptin	Amlodipine, Losartan, Acetylsalicylic acid, Magnesium hydroxide, Thioctic acid	7.7	41.4	Neuropathy, retinopathy
16	148	72	32.8	70	7	Metformin, Gliclazide	Atorvastatin, Amlodipine, Acetylsalicylicacid, Magnesiumhydroxide, Allopurinol	9.9	29.4	Neuropathy, retinopathy
17	172	105	35.5	64	3	Metformin, Glimepiride, Vildagliptin	Losartan, Hydrochlorothiazide, Atorvastatin, Acetylsalicylicacid	7.6	53.4	Neuropathy, retinopathy
18	160	76	29.7	70	12	Metformin, Liraglutide	Valsartan, Verapamil, Indapamide, Thioctic acid	8.2	50.2	Neuropathy, retinopathy
19	159	144	45	46	18	Metformin, Vildagliptin, Gliclazide	Losartan, Amlodipine, Moxonidine, Indapamide, Acetylsalicylicacid	10.2	34.4	Neuropathy, retinopathy
20	164	72.1	26.8	74	10	Metformin, Glibenclamide	Amlodipine, Bisoprolol, Atorvastatin, Acetylsalicylic acid	9.3	49	Neuropathy

**Table 5 ijms-20-02329-t005:** Physical and biochemical parameters of individuals without diabetes, as well as corresponding therapy, not related to diabetes.

Participants	Height (cm)	Weight (kg)	Body Mass Index	Age (Years)	Therapy		HbA_1C_ (%)	Content of Albumin in Blood (μg/μL)	Complications of Type 2 Diabetes Mellitus
					Diabetes-Specific Therapy	Diabetes-Nonspecific Therapy			
1	162	78	29.7	54	-	Enalapril	5.8	47.3	-
2	156	82	33.7	67	-	-	6.2	46	-
3	168	84	29.8	67	-	-	5.7	47.8	-
4	171	77	26.6	57	-	Acetylsalicylic acid	5.4	49.2	-
5	159	83	32.8	57	-	Amlodipine, Atorvastatin	5.5	53.4	-
6	158	75	30.0	65	-	Losartan, Indapamide, Metoprolol, Acetylsalicylic acid, Atorvastatin	6.0	46.4	-
7	158	64.1	25.7	64	-	-	4.9	45.9	-
8	163	83	31.2	61	-	Losartan, Acetylsalicylic acid, Calcium carbonate, Colecalciferol, Alfacalcidol	5.1	39.8	-
9	158	64	25.6	64	-	Enalapril	5.2	50	-
10	173	67	22.4	58	-	-	5.5	40.1	-
11	163	72	27.1	61	-	-	5.7	48.4	-
12	165	91.8	33.7	55	-	Amlodipine, Valsartan, Hydrochlorothiazide	5.3	49	-
13	154	59	24.9	57	-	Atorvastatin, Acetylsalicylicacid, Magnesiumhydroxide, Indapamide, Calciumcarbonate, Colecalciferol	5.4	49.9	-
14	166	70	25.4	56	-	Bisoprolol, Red vine leafextract	5.9	43.2	-
15	162	85	32.4	63	-	Losartan, Hydrochlorothiazide	5.8	49	-
16	164	78	29.0	55	-	Losartan, Hydrochlorothiazide	5.0	50.2	-
17	172	95	32.1	64	-	Perindopril, IndapamideAmlidipine	5.1	39	-
18	155	72.5	29.8	68	-	Rosuvastatin, Perindopril, Indapamide, Levothyroxine, Calcium carbonate, Colecalciferol, Ibandronic acid	5.6	48.2	-

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
