# Peer review of "Multiple Glycation Sites in Blood Plasma Proteins as an Integrated Biomarker of Type 2 Diabetes Mellitus"

_ijms, 2019, doi:10.3390/ijms20092329_

Reviewer 1 Report

The authors present a rather complex manuscript focusing on the validation of a biomarker focusing on multiple glycation sites in plasma protein

My report focuses on the general topic of glycation and biomarkers of T2DM, rather than the proteomic angle, which I did not feel was presented in a very accessible way.

My main concern is the number of case and controls used in this paper, and the lack of description of these people in the main text (including disease duration).

There are no comparisons between the new biomarker and existing ones, and the value of it is not demonstrated clearly.

The statistics section is lacking details, and no sample size calculation is provided.

introduction  - it is not clear how K141 in haptoglobin would provide an additional diagnostic tool. Why is it necessary or useful, beyond established markers?

on page 3, I cannot follow the logic of the statement "In this context, it is logical to assume, that a biomarker strategy, based on multiple specific

glycation sites, could essentially increase the efficiency of glycemic control and disease prediction."

How is the current biomarker strategy, based on HbA1c and FBG failing, and where are the sites mentioned in the statement above?

The paradigm explored by the manuscript only considers glucose as a driver for glycation - in fact, oxidative stress has been shown to be another important driver, and glycation can be modulated by dietary bioactives, including polyphenolics.

Throughout - "diabetic" should not be used as an adjective - people with diabetes is the accepted form

In results 2.1 - I do not understand the inter-gel normalisation procedure.

Table 1: it is not clear which peptides were identified in people with / without T2DM. It is not clear what was tested using the MW U-test.

Author Response

We thank the reviewer for the thoughtful review and highly appreciate the valuable comments and suggestions to improve the manuscript. Following these advices we performed all required changes in corresponding sections, as indicated in the following rebuttal addressing each aspect.

Reviewer 1

Comments and Suggestions for Authors

Major remark 1:“My main concern is the number of case and controls used in this paper, and the lack of description of these people in the main text (including disease duration).”

Answer: We agree with the reviewer and realize the importance of the addressed issues. Here we address each of them one by one:

1)        Number of case and control samples:

We agree with the reviewer – we did not provide any information about our rationales behind the cohort sizes selected. The main reason for this was that this was a pilot study, and we had no source to estimate possible effect sizes (and thus no reliable pre-calculation of sample size could be done before the experiments). Although the number of samples for the Mann-Whitney test could be higher, but, to our view, it was sufficient. Due to high amount of potential marker peptides and since we were interested in peptides which would have their Mann-Whitney test’s H0 rejected, in order to prevent ‘p-level hacking’ (Douglas Curran-Everett, 2000, Am J Physiol Regul Integr Comp Physiol. Jul;279(1):R1-8.), in our analysis, α was set to 0.01 (instead of typical 0.05) and specific multiple comparison correction procedure (Holm, S., 1979, Scand. J. Stat., 6, 65–70.) was applied to ensure family-wise error rate is retained at proper level. As far as PCA\LDA is concerned, for these methods, dimensionality reduction techniques are well suited for high parameters of samples ratio, and according to Sharma et al (2015. Int. J. Mach. Learn. & Cyber. 6: 443) and references therein, our case (approx 40 dimensions to 40 samples) is by several orders of magnitude below typical thresholds when small sample size problem should be specifically addressed.

The corresponding text is added to discussion:

“This cohort size was a good starting point for this pilot study, but due to a high amount of potential marker peptides we employed lower than usual α-level (0.01) and implemented family-wise error rate control procedure [] , thus ensuring that samples sizes used in our research are sufficient to prove all related inferences.As for PCA\LDA. For PCA\LDA, dimensionality reduction techniques are well suited for high parameters of samples ratio, and according to Sharma et al [ ] and references therein, our case (approx 40 dimensions to 40 samples) is by several orders of magnitude below typical thresholds when small sample size problem should be specifically addressed” (lines 251-257).

2)        Patient description:

The tables with description of case and controls were moved into the main text as requested.(lines 448 and 449)

3)   Disease duration:

This information is provided as requested by the reviewer. Thus, we added information about disease duration in table Table 4“Physical and biochemical parameters of the T2DM patients, as well as corresponding diabetes-related and non-related therapy.” (line 448)

Major remark 2:“There are no comparisons between the new biomarker and existing ones, and the value of it is not demonstrated clearly.”

Answer:We agree with the reviewer – this information would be helpful. Therefore, in discussion we provide comparison of our new biomarker with the existing one – glycated hemoglobin, and provide additional figure (Figure 7). The corresponding text is provided in text:

“i.e. provide separation of groups not worth than existing ones, e.g. glycated hemoglobin (Figure 7). However, the biomarker, based on multiple glycated proteins would give a higher degree in flexibility of glycemic control depth.” (lines 332 – 334, 344-347)

Major remark 3: The statistics section is lacking details, and no sample size calculation is provided.”

Answer: We agree with the reviewer – more details on statistic need to be provided in discussion (in the methods part it looks sufficient). It is important to mention, that "sample size calculation" is not applicable here, as it is a pilot study. In this case, analysis of statistical power (done post-assay) is applicable. In agreement with this, changes in text are done as described in the answer to the Major critique 1.(lines 251-257)

Major remark 4:Introduction  - it is not clear how K141 in haptoglobin would provide an additional diagnostic tool. Why is it necessary or useful, beyond established markers?

Answer: We complemented text with additional information about K141.

„The main advantage of combining two markers - HbA1c and glycated haptoglobin K141 is simultaneous consideration of two proteins with different half-life times, i.e. 3–4 and 2 - 4 days, respectively. It makes this biomarker sensitive to long- and short-term fluctuations of blood glucose concentrations. The set of glycated K141 of haptoglobin and HbA1c provided a sensitivity of 94 %, a specificity of 98 %, and an accuracy of 96 % to identify T2DM.” (Lines 88-93)

Major remark 5:“on page 3, I cannot follow the logic of the statement "In this context, it is logical to assume, that a biomarker strategy, based on multiple specific glycation sites, could essentially increase the efficiency of glycemic control and disease prediction."”

Answer: The answer is provided in the text:

“Involvement of several glycated proteins with different half-life times (τ1/2) allows addressing several time segments in glycemic control without additional analyzes.” (lines 97-98)

Major remark 6:“How is the current biomarker strategy, based on HbA1c and FBG failing, and where are the sites mentioned in the statement above?”

Answer: We think that the fasting plasma glucose (FPG) is the preferred diagnostic parameter but the increasing of blood glucose detected also in patients, suffering from other, besides diabetes, diseases.HbA1c reflects the average plasma glucose level for the preceding three-month period and correlates well with micro- and macrovascular сomplications.HbA1c does not deliver any information about short-term alterations in plasma glucose concentrations, accompanying onset of metabolic syndrome (lines 61-66). All glycated peptides identified during our experiment belonged to the proteins with varying half-life times, we succeeded to address a time scale of glycemic control. Indeed, relatively long-living proteins (HSA and Ig kappa chain C region with τ1/2 of up to three weeks) are exposed to plasma glucose for the times much longer than those relevant for α-2-macroglobulin and complement C4-A protein (τ1/2 of 1 – 3 days). Hence, the glycation sites representing the latter proteins would deliver valuable information about the blood sugar levels over the days directly preceding the analysis. Such data would directly show if short-term fluctuations of plasma glucose levels do occur. (lines 321-323):

“Hence, the glycation sites representing the latter proteins would deliver valuable information about the blood sugar levels over the days directly preceding the analysis. Such data would directly show if short-term fluctuations of plasma glucose levels do occur”

We mentioned “glycation sites” in light of “multiple specific glycation sites in plasma proteins.”(line 96)

Major remark 7: The paradigm explored by the manuscript only considers glucose as a driver for glycation - in fact, oxidative stress has been shown to be another important driver, and glycation can be modulated by dietary bioactives, including polyphenolics.

Answer: We completely agree with the Reviewer that polyphenols play important role in diabetes course. Polyphenols decrease hyperglycemia and improve acute insulin secretion and insulin sensitivity. The corresponding considerations are appended to the text.

‘Also in future for collecting blood samples, it is necessary to keep in mind not only therapy, but also the diet of patients. Thus, dietary polyphenols influenced on blood glucose at different levels and may also help control and prevent diabetes complication via decrease of hyperglycemia and improvement of acute insulin secretion and insulin sensitivity.” (lines 367-370)

Major remark 8:Throughout - "diabetic" should not be used as an adjective - people with diabetes is the accepted form.”

Answer: Corrected accordingly

(lines 103-104, 110, 136-137, 139,143, 157, 181, 184, 251, 324, 411, 449, 519, 524)

Majorremark 9:“In results 2.1 - I do not understand the inter-gel normalisation procedure.”

Answer:The aim of the normalization procedure is to exclude any effects of methodological bias, related to errors in sample preparation and especially, determination of protein concentrations. Therefore, after separation of the proteins, the total density of all bands in each sample were calculated and inter-sample comparisons were done. We added information aboutinter-gel normalization in part “Material and methods, 5.3 Blood sampling and plasma isolation”:

“Average densities across individual lanes (expressed in arbitrary units) were determined by ChemiDoc XRS imaging system controlled by Quantity One® 1-D analysis software (Bio-Rad Laboratories Ltd., Moscow, Russia). Thereby, for inter-gel normalization, the first and the last plasma protein samples loaded on each gel were replicated in the previous and following gels, respectively. For calculation of RSDs, the densities of individual lines were normalized to the gel average value.” (lines 425-430)

Major remark 10:Table 1: it is not clear which peptides were identified in people with / without T2DM. It is not clear what was tested using the MW U-test.”

Answer: In Table 1all glycated peptides identified in pooled plasma samples from T2DM and healthy patients are summarized. Our study revealed that protein glycation is present already in group of healthy individuals, but the levels of glycation relatively were lower.

This is reflected in text:

“This strategy revealed totally 51 Amadori-modified peptides present in pooled plasma protein digests in each sample and representing ten major plasma proteins.” (lines 135-137)

Mann-Whitney test compared integrated peak areas of glycated peptides. For each glycated peptide, peak areas were integrated across the whole dataset. During Mann-Whitney test, two groups were compared: one group included peak areas obtained from samples from patients with diabetes and other group - from normoglycemic controls.

“As these features could be reliably detected, i.e. demonstrated a signal to noise ratio ≥ 3 in corresponding XICs, their relative abundances in T2DM and without diabetes cohorts were addressed by the label free quantification approach.”(lines 137-140)

 “Paired Mann-Whitney test performed for the intensities of individual peptide signals (expressed as integral peak areas of corresponding XICs at defined retention times (tRs).” (lines 150-152).

“Statistical significance of differences in the relative abundances of specific glycated peptides detected in the plasma protein digests obtained from T2DM patients and healthy individuals was determined by the Mann-Whitney U test.” (lines 518-520)

Reviewer 2 Report

In introduction section page 2 (paragraph 1), author should add more information about HSA glycation and cite article “Int J Biol Macromol. 2019, 123: 979-990”.

Author Response

We cite the article „Int J Biol Macromol. 2019, 123: 979-990“.

“The levels of HSA glycation varied from 1 % and 10 % in healthy persons till 20-90 % in patients with” diabetes. (lines 72-73)